# Discovery of Adversarial Endgame Chess Positions

## ABSTRACT

Chess engines have become an essential component of today's lucrative online chess market, and many players treat their recommendations as the ground truth. However, these engines are not perfect and can make mistakes when faced with certain endgame positions. The occurrence of such positions within an engine's search could lead to errors cascading to the root. Despite this, the systematic generation and analysis of positions that expose such weaknesses remains an underexplored area of research. To fill this gap, we develop **ADVCHESS**, a novel framework to automatically generate adversarial chess positions. These are positions where state-of-the-art engines deviate from theoretically optimal play. Our approach focuses on identifying fair and legal positions where engine failures result in significant outcome changes, particularly in the context of endgame play, where ground-truth labels can be extracted from specialized endgame tablebases. We design state and action encodings as well as a reward function for the foundation of the generative modeling problem. We find that adversarial positions generated for Stockfish are least transferable across different computational settings and that transferability does not correlate directly with engine strength.

## 1 INTRODUCTION

Chess has long served as a benchmark domain for advances in artificial intelligence (AI), from early game tree search algorithms to contemporary deep learning applications. Today, chess represents a significant professional domain with growing tournament prize pools, substantial corporate sponsorship, and a thriving online ecosystem. Modern chess engines such as Stockfish (Romstad et al., 2024), Leela Chess Zero (Pascutto & Linscott, 2024), and AlphaZero (Schrittwieser et al., 2020) have consistently outperformed human players in the past two decades. This success comes from continuous algorithmic improvements, active open-source development communities, and increased computational resources.

Despite significant advances in computer chess, modern engines still make mistakes in certain endgame positions (Sadmine et al., 2023). We refer to these configurations as *adversarial positions*: specialized board configurations that reveal weaknesses in engine decision making, leading to suboptimal play and altering theoretical game outcomes from wins to draws or losses. The systematic generation and analysis of such positions remains an underexplored area of research.

The automatic discovery of adversarial positions is difficult for several reasons. First, the sheer number of possible chess positions, even in endgame scenarios, makes exhaustive search impractical. Second, given a specified chess position, we need to determine whether it is, in fact, an adversarial position. Finally, while tablebases provide perfect play information for positions with up to 7 pieces, systematically leveraging this knowledge to generate positions that expose engine weaknesses remains a complex optimization problem.

Prior research in chess engine analysis and position generation differs from our objective of systematically generating positions that expose engine gameplay vulnerabilities. A recent work (Sadmine et al., 2023) identified critical weaknesses in top chess engines' endgame play; however, they rely on uniform position generation, constraining them to subsets of five-piece EGTBs. Similarly, systematic evaluations of Leela's endgame performance (Haque et al., 2021) and studies of engine behavior in Chinese checkers using AlphaZero-style agents (Karki, 2024) analyze existing positions rather than generating adversarial ones. In the domain of position generation, there exists a single recent work (Pettersson, 2024), which employed deep generative models, specifically generative adversarial

networks and decoder-only transformers. In this work, the objective was to generate realistic chess positions, rather than those that induce suboptimal play. Overall, none of these approaches address the automated discovery of positions to systematically expose chess engine vulnerabilities.

In this paper, we introduce **ADVCHESS**, a framework for the systematic discovery of adversarial chess positions. Within the iterative process, positions are evaluated based on outputs from the target chess engine, EGTBs, and other heuristics. Since the discovery of adversarial chess positions can be seen as a sampling problem from a complex and unknown distribution of game states, the heart of this framework is an interchangeable sampling algorithm. The performance of the sampling algorithm determines the performance of the entire system.

To find an effective sampling algorithm beyond the uniform sampling methods used to generate chess positions in prior works, we first benchmark the effectiveness of several state-of-the-art sampling algorithms, including uniform sampling, Markov Chain Monte Carlo (MCMC) (Metropolis et al., 1953), Generative Flow Networks (GFlowNets) (Bengio et al., 2021) and a generative policy trained via Proximal Policy Optimization (PPO) (Schulman et al., 2017b).

Our experiments demonstrate that uniform sampling is ineffective in finding adversarial positions due to a lack of guidance in the search. To apply reward-based methods to this task, we introduce a novel reward function that takes into account the adversarial score, the legality of the positions (whether it is valid according to the rules of chess), and the material balance (how many pieces each side has) of the generated position. However, PPO, a reward based method, suffers from a lack of diversity in the generated positions due to mode collapse. The Metropolis-Hastings algorithm (a type of MCMC) is able to generate a wide variety of adversarial positions while also maintaining their legality and material balance. GFlowNets are unable to outperform Metropolis-Hastings in this task, unlike their reported success on different problems such as molecular generation (Bengio et al., 2021) and prompt engineering (Lee et al., 2024).

Upon closer investigation of the positions generated by Metropolis-Hastings, we observe that it exploits a crucial property that GFlowNets cannot: adversarial positions are densely clustered in the input domain. An intuition for this can be found in Subsection 4.3 and Figure 3. Based on these insights, we design a novel search algorithm, **Adversarial Search via Local Exploration (AS-LE)**, which leverages a combination of local and uniform search to effectively explore the vast space of chess positions. This method significantly outperforms all previous approaches. In summary, our key contributions are as follows:

- We benchmark several advanced sampling methods (MCMC, PPO, and GFlowNets) against the uniform baseline, and analyze their performance and limitations in adversarial position discovery. This includes the design of a reward function with adversarial score, legality, and material balance and suitable state and action spaces.
- Through our benchmarks, we gain the crucial insight that adversarial chess positions are densely clustered. Based on this, we design and implement a novel search algorithm, AS-LE, as the core sampling component within our **ADVCHESS** framework, which combines local and uniform search and significantly outperforms all tested baselines, finding between $2.2\times$ and $10.2\times$ more adversarial samples than the best baseline.
- We evaluate our framework by benchmarking it against three diverse chess engines (Stockfish, Winter, and Floyd) with three different metrics. We measure intra-engine transferability to be as low as just 12%. Inter-engine transferability ranges between 7% and 63%.

## 2 PRELIMINARIES

**Chess Engines and Endgame Tablebases.** Modern chess engines, such as Stockfish (Romstad et al., 2024), achieve superhuman performance by combining deep search algorithms with powerful position evaluation functions. The search, typically a highly optimized variant of Alpha-Beta pruning (Knuth & Moore, 1975), explores a vast tree of move sequences. The *evaluation function* assigns a score to board positions within this tree, predicting the game's outcome.

In the endgame (positions with few pieces), engine evaluations can be inaccurate (Sadmine et al., 2023). This weakness is perfectly solved by *Endgame Tablebases (EGTBs)* (Ströhlein, 1970), which are precomputed databases containing the exact win/loss/draw outcome for every possible position up

to a certain number of pieces. However, their large size (the six-piece EGTB ranges from 150GB to 1.2TB depending on the format) makes them impractical for resource-constrained environments, such as chess engines running in a Web browser. In such cases, engines must fall back on their potentially flawed search and evaluation.

**Sampling Methods.** The discovery of adversarial chess positions can be seen as a *sampling* problem from a complex and unknown distribution of game states. Finding successful outputs in this manner is useful in many fields, from designing new molecules (Bengio et al., 2021) to creating jailbreak prompts for large language models (Lee et al., 2024). In chess, prior work has relied on uniform (sometimes filtered) random sampling (from datasets (Lai, 2015) or from scratch (Sadmine et al., 2023)) or from simulated self-play (Schrittwieser et al., 2020). To navigate the vast state space and find adversarial positions, we investigate integrating four different sampling methods into **ADVCHESS**: Uniform Sampling, Markov Chain Monte Carlo, Generative Flow Networks, and a generative policy finetuned with Proximal Policy Optimization. We also introduce our own sampling algorithm, AS-LE, with all of these components detailed in Section 4.

*Markov Chain Monte Carlo (MCMC)* methods sample from a probability distribution by constructing a Markov chain that has the desired distribution as its equilibrium distribution. We use the Metropolis-Hastings algorithm (Metropolis et al., 1953; Hastings, 1970), a prominent MCMC technique that accepts or rejects proposed states to explore the distribution.

*Proximal Policy Optimization (PPO)* (Schulman et al., 2017b) is a widely adopted policy gradient algorithm known for its robustness and sample efficiency. We use PPO, a reinforcement learning method, to train a policy that sequentially constructs chess positions based on the reward signal.

*Generative Flow Networks (GFlowNets)* (Bengio et al., 2023) are a more recent class of probabilistic generative models. They learn a policy to sequentially construct an object $x$ such that the probability of generating it, $P(x)$, is proportional to a given non-negative reward function $R(x)$. While many RL algorithms, such as PPO, aim to find a single policy or action that maximizes the expected reward (Sutton & Barto, 2018), GFlowNets are designed to learn a stochastic policy that samples a diversity of high-reward outcomes.

## 3 PROBLEM STATEMENT AND MOTIVATION

Let $\mathcal{B}$ be the set of all valid chessboard positions, $\mathcal{T} \subset \mathcal{B}$ be the set of all terminal positions such as checkmate, stalemate, and draw by rule, and $\mathcal{M}(b)$ be the set of all valid moves for each position $b \in \mathcal{B}$. In addition, let $g : b \in \mathcal{B} \times \mathcal{M}(b) \to \mathcal{B}$ be the deterministic transition kernel that applies a move to a position resulting in a new position and let $h : \mathcal{T} \to \{-1, 0, 1\}$ be a function that maps each terminal state to a result, where, for the player with the white piece, $-1$ represents a loss, $1$ a win, and $0$ a draw.

We represent the chess engine as a function $f : b \in \mathcal{B} \to \mathcal{M}(b)$ that maps each position to a corresponding *perceived* optimal valid move. Therefore, the chess engine's input is the current position of the board, and its output is a single move that the engine believes to be the best in the current position.

An EGTB $t : b \in \mathcal{B} \to \mathcal{M}(b)$ is similar to a chess engine, with the only difference being that it is the ground truth database that maps each board position to the corresponding *optimal* move[1]. Here, when $f(b) \neq t(b)$ for some $b \in \mathcal{B}$, the chess engine $f$ might have made a mistake.

We introduce the perfect-play operator $\mathbb{O} : \mathcal{B} \to \{-1, 0, 1\}$ to map each position to the corresponding result under perfect-play (where both players make optimal moves). We define $\mathbb{O}$ recursively as

$$\mathbb{O}(b) = \begin{cases} \mathbb{O} \circ g(b, t(b)) & b \notin \mathcal{T} \\ h(b) & b \in \mathcal{T} \end{cases} \tag{1}$$

Given a target chess engine $f$, our objective is to generate a set of positions $\mathcal{A} \subseteq \mathcal{B}$ such that following the engine's move immediately and thereafter playing optimally leads to a result different from the one where all moves are optimal. Formally, we aim to generate positions $b \in \mathcal{A}$ where

$$\mathbb{O} \circ g(b, f(b)) \neq \mathbb{O}(b) \tag{2}$$

---

[1]This is simplified for ease of notation. EGTB returns outcomes from which optimal moves can be derived.

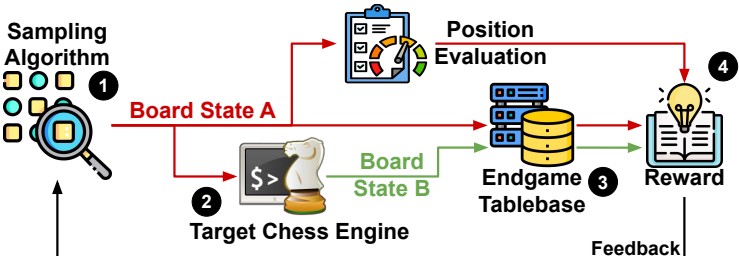

Figure 2: Overview of **ADVCHESS**.

We present two examples, 5-piece and 6-piece, from our evaluation, in which Stockfish (Romstad et al., 2024), the strongest chess engine available, makes mistakes. These examples highlight the existence of adversarial positions.

Figure 1a illustrates a position where Stockfish, within a $400$ search-node query budget (i.e. the engine is allowed to evaluate 400 positions within its search), fails to make the winning move of capturing a free bishop (red arrow). Instead, it blunders by retreating its knight (green arrow), allowing the opponent to escape into a draw from a losing position. In Figure 1b, Stockfish fails to find the optimal move, even with a $64,000$ search-node budget. Instead of capturing the unprotected knight with its queen to win (red arrow), it sacrifices its queen with a king move (green arrow), leading to a draw.

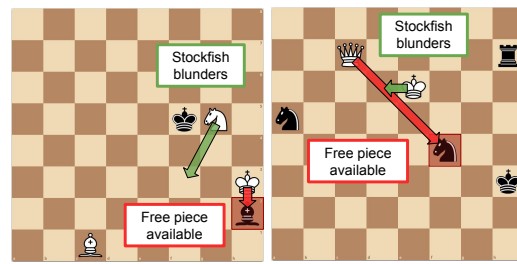

(a) 5 Piece Stockfish Error  (b) 6 Piece Stockfish Error

Figure 1: Illustration of Adversarial Chess Positions. Average human players would capture the free pieces respectively (in red), achieving a winning position. Stockfish, makes a critical error in these cases (in green).

Our goal is to generate positions that satisfy Equation 2. However, there are several challenges that must be addressed:

**Sparsity of Adversarial Positions ($C_1$).** The search space of all legal chess positions is vast; e.g., a 6-piece EGTB contains $3.8 \times 10^{12}$ legal positions (Kryukov, 2025). Yet, only a small fraction of these induces mistakes in state-of-the-art engines (See Appendix Table 1).

**Complexity of Strategies ($C_2$).** A high-level game of chess involves long-term strategic dependencies. This complex interplay of rules and strategy complicates the search for adversarial positions. For instance, a seemingly minor pawn move can have significant future consequences, such as opening lines for other pieces or creating a passed pawn that could promote to a more powerful piece. These long-term effects make it challenging to identify adversarial positions.

**Legality and Balance Constraints ($C_3$).** Generated positions must satisfy both chess rules and practical gameplay considerations. Legality constraints encode chess rules (e.g., no pawns on back ranks, one king each side) while balance constraints enable material balance (since heavily biased positions do not occur in practice).

**Diversity of Generated Positions ($C_4$).** Generating a diverse set of adversarial positions is crucial for comprehensive engine testing and improvement. However, optimization-based approaches often converge to a limited set of similar positions that reliably induce engine mistakes, a phenomenon analogous to mode collapse in generative models. This lack of diversity can lead to overfitting and result in failure to reveal the full spectrum of adversarial positions.

## 4 DISCOVERING ADVERSARIAL POSITIONS

We introduce **ADVCHESS**, an iterative framework to find rare, failure-inducing endgame positions by navigating the vast and sparse search space. Figure 2 shows its four components:

A *sampling algorithm* generates a candidate chess position $b \in \mathcal{B}$ ❶. We implement and evaluate several algorithms, including a novel search technique designed specifically for this problem, to explore the vast state space efficiently. The generated position $b$ is presented to a target chess engine $f$, which determines its perceived best move within a fixed compute budget ❷. We use an Endgame Tablebase (EGTB) to determine the theoretical ground-truth outcome of the position before and after the engine's move ❸. The EGTB provides the perfect-play operator $\mathbb{O}$, which maps any position to its true outcome (win, draw, or loss). We check if the position is adversarial by testing if the engine's move altered the game's theoretical outcome, i.e., if $\mathbb{O} \circ g(b, f(b)) \neq \mathbb{O}(b)$. If so, the position is stored. Concurrently, a **reward signal** $R(b)$ is calculated based on the outcome, legality, and material balance of the position ❹. This reward provides the feedback signal for our learning-based samplers. The sampling algorithm uses the calculated reward $R(b)$ to update its internal parameters. This guides subsequent searches toward more promising state space regions. It then returns to Step ❶.

## 4.1 SAMPLING ALGORITHM ADAPTATION

To discover adversarial positions, there is a set of techniques that can be used. To solve this problem, prior work relied on a uniform sampling approach (Sadmine et al., 2023). In this approach, a position is generated by first placing the white and black kings on two distinct, uniformly random squares. Subsequently, the remaining non-king pieces are selected uniformly at random and placed on uniformly random empty squares until the desired total number of pieces is on the board. However, uniform sampling does not address challenges $C_1$, $C_2$, and $C_3$. This is because it is agnostic towards the objective, as evidenced by the results in Section 6.

Challenges $C_1$ and $C_3$ can be mitigated with algorithms guided by a reward function, including the Metropolis-Hastings algorithm (an MCMC method) (Metropolis et al., 1953; Hastings, 1970), Proximal Policy Optimization (PPO) (Schulman et al., 2017a), and Generative Flow Networks (GFlowNet) (Bengio et al., 2023). However, these more advanced techniques are not a complete solution, as they fail to address all challenges. Reinforcement learning methods like PPO, for instance, are notoriously susceptible to mode collapse, a phenomenon that severely limits the diversity of generated samples (Challenge $C_4$).

MCMC requires the definition of a Markov chain with appropriate proposal distributions and a target distribution. GFlowNets and PPO require defining state and action encodings and a suitable reward function that captures the legality and balance of a position while also rewarding adversarial positions. We address these requirements for each algorithm as follows.

**MCMC.** We define the initial state as a position that contains two kings and a pre-defined number of other pieces, all placed uniformly at random. At each step, we propose a new candidate position by either moving a random piece to a random empty square or by transforming a random non-king piece and moving it to a random empty square. The candidate position is accepted with a probability based on the ratio of rewards, $\alpha = \min(1, R(b')/R(b))$, allowing the search to explore the reward landscape effectively.

**PPO and GFlowNet.** For these learning-based methods, we frame the generation as a sequential process. Starting with a legal, random placement of the two kings, the agent's policy is used to sequentially add the remaining non-king pieces to the board. The action space is a discrete distribution of size $64 \times 10$, representing the placement of one of ten possible pieces (White/Black Pawn, Knight, Bishop, Rook, Queen) on one of the $64$ squares.

The above algorithms require the definition of a target probability distribution (also called the reward). We design the reward to simultaneously encode three desired aspects (1) adversarial score, (2) position legality, and (3) material balance of the position, detailed below.

## 4.2 REWARD FUNCTION FOR DISCOVERY

A well-designed reward function is critical for guiding the generative process toward our objective. Our function $R(b)$ translates the formal goal from Equation 2 and the associated constraints into a computable score for a given position $b$. It the sum of four distinct components: the *Baseline Reward* $\beta_b$, the *Constraint-Handling Rewards* $R_l(b)$ and $R_m(b)$, and the *Adversarial Outcome Reward* $R_o(b)$.

**Adversarial Outcome Reward** ($R_o$). This is the primary component, directly rewarding the discovery of adversarial positions. It is non-zero only if an engine's move $f(b)$ leads to a state with a different theoretical outcome. Using the perfect-play operator $\mathbb{O}$, we define it as:

$$R_o(b) = \beta_o \cdot \left(\mathbb{O}(g(b, f(b))) - \mathbb{O}(b)\right)^2 \tag{3}$$

This outcome-centric reward directly measures engine failure, circumventing the need to analyze complex move sequences ($C_2$).

**Constraint-Handling Rewards.** We introduce two rewards to ensure legal and balanced positions: *Valid Position Reward ($R_l$):* A fixed reward $\beta_l$ is given if the position $b$ is legal according to the rules of chess (e.g., no pawns on back ranks, correct number of kings). This steers the search away from invalid states.

*Balanced Material Reward ($R_m$):* A reward $\beta_m$ is given if the material difference between the two players is within a small, predefined threshold. This encourages the generation of plausible endgame positions and is only added if the position is also legal.

**Baseline Reward** ($\beta_b$). A small, constant baseline reward $\beta_b$ is added to every position. This enhances numerical stability during training and encourages wider exploration, which helps to counteract the sparsity of adversarial positions ($C_1$). For the parameters used in our experiments, refer to Appendix B.

### 4.3 ADVERSARIAL SEARCH VIA LOCAL EXPLORATION

Based on results from prior work (Bengio et al., 2021), we expect GFlowNet to outperform MCMC. However, in our problem space, MCMC outperformed GFlowNets (See Section 6). Upon investigation of the adversarial positions generated by the MCMC algorithm, we find that it benefits from adversarial positions that are densely clustered in the Markov chain, meaning if we have some adversarial position $x_t$, then a minor perturbation $x_t'$ has a high probability of also being adversarial. Figure 3 shows an example and explanation of this phenomenon. If an engine were to not see the checkmate, the sampling algorithm could move the pawn in each of the two examples to any square within the red box without impacting the position too greatly, resulting in more adversarial positions.

Based on this observation, we develop a new sampling algorithm called Adversarial Search via Local Exploration (AS-LE), which combines uniform random sampling with a local search component. Our results indicate that in our domain this method outperforms the more well-established and general methods (See Section 6).

AS-LE operates in two phases. In the **Discovery** phase, it begins with a uniform random search until an initial adversarial position is found. In the **Exploration** phase, upon finding a seed, it initiates a Depth-First Search (DFS) from that position to explore its local neighborhood. A "neighbor" is generated by moving a single piece to any empty square. This allows for broad ex-

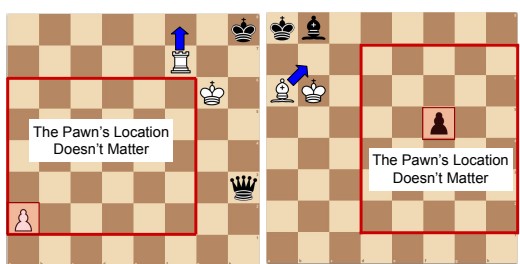

Figure 3: If an engine is not able to find the checkmate (Blue) then any pawn in the red box would also likely be an adversarial position.

ploration of similar board configurations. To maintain diversity in piece configurations, there is a chance (hyperparameter) that any non-king piece being moved is transformed into a different random piece type.

To manage the search, the algorithm maintains a record of visited positions to avoid redundant exploration. If the search stack grows beyond a certain size (e.g., 10,000), half of it is pruned to manage memory. If the search stack ever runs out due to finding too few adversarial positions, the algorithm returns to the Discovery Phase.

# 5 EXPERIMENTAL SETUP

We describe the implementation of **ADVCHESS**, the chess engines used for evaluation, and the metrics defined to benchmark sampling algorithms. Further implementation details and algorithm hyperparameters are given in the Appendix B.

## 5.1 CHESS ENGINES

We evaluate **ADVCHESS** using three chess engines: Stockfish dev-20241004-81c1d310 (Romstad et al., 2024), Winter v4.02 (Rosenthal, 2024), and Floyd v0.9 (Kervinck, 2016), selected to represent a range of evaluation functions and playing strengths. All use variants of the Alpha-Beta search. We query each engine in a single-threaded fashion with a 1 MB transposition table and with fixed search-node budget of 400, 800, 1600, 3200, and 6400 nodes. A higher search-node budget implies longer computation time and a more accurate evaluation.

## 5.2 EVALUATION METRICS

**Unique Adversarial Positions (UAP).** UAP measures the number of *unique* adversarial positions generated after a given number of training episodes. Let $\mathcal{P}_c$ be the set of positions generated for a chess engine $c$, $g$ be the transition kernel that represents moves, and $\mathbb{I}_A^c(p)$ be an indicator function for an adversarial position $p \in \mathcal{P}_c$, defined as:

$$\mathbb{I}_A^c(p) = \begin{cases} 1 & \text{if } \mathbb{O} \circ g(p, c(p)) \neq \mathbb{O}(p) \\ 0 & \text{otherwise} \end{cases}$$

The UAP is then defined as:

$$\text{UAP}(\mathcal{P}_c) = \sum_{p \in \mathcal{P}_c} \mathbb{I}_A^c(p)$$

**Unique Success Rate (USR).** USR is the percentage probability that the next generated position is a *new* unique adversarial position. Let $\mathcal{P}_c^i$ be the set of the first $i$ positions generated for a chess engine $c$.

$$\text{USR}\left(\mathcal{P}_c^i\right) = \text{P}\left(\text{UAP}\left(\mathcal{P}_c^{i+1}\right) > \text{UAP}\left(\mathcal{P}_c^i\right)\right) \times 100$$

Since we cannot compute the USR exactly for a single sample during training, we estimate it over a batch of size $j$:

$$\text{USR}\left(\mathcal{P}_c^i\right) \approx \frac{\left(\text{UAP}\left(\mathcal{P}_c^{i+0.5 \times j}\right) - \text{UAP}\left(\mathcal{P}_c^{i-0.5 \times j}\right)\right)}{j} \times 100$$

**Adversarial Position Transferability (APT).** Let $c_1$ and $c_2$ be two different chess engines or the same engine with different settings. Let $\mathcal{A}_{c_1}$ be the set of adversarial positions generated for $c_1$. The Adversarial Position Transferability, $\text{APT}(c_1, c_2)$, is the percentage of positions in $\mathcal{A}_{c_1}$ that are also adversarial positions for $c_2$. Formally, it is defined as:

$$\text{APT}(c_1, c_2) = \frac{|\{p \in \mathcal{A}_{c_1} \mid \mathbb{I}_A^{c_2}(p) = 1\}|}{|\mathcal{A}_{c_1}|} \times 100$$

# 6 EVALUATION RESULTS

This section evaluates the performance of **ADVCHESS**. We first compare our proposed sampling algorithm, Adversarial Search via Local Exploration (AS-LE), against four other methods to establish its superior efficiency. We then analyze the performance of AS-LE in generating adversarial positions for different chess engines and computational budgets. Finally, we assess the transferability of the discovered positions across various engine configurations.

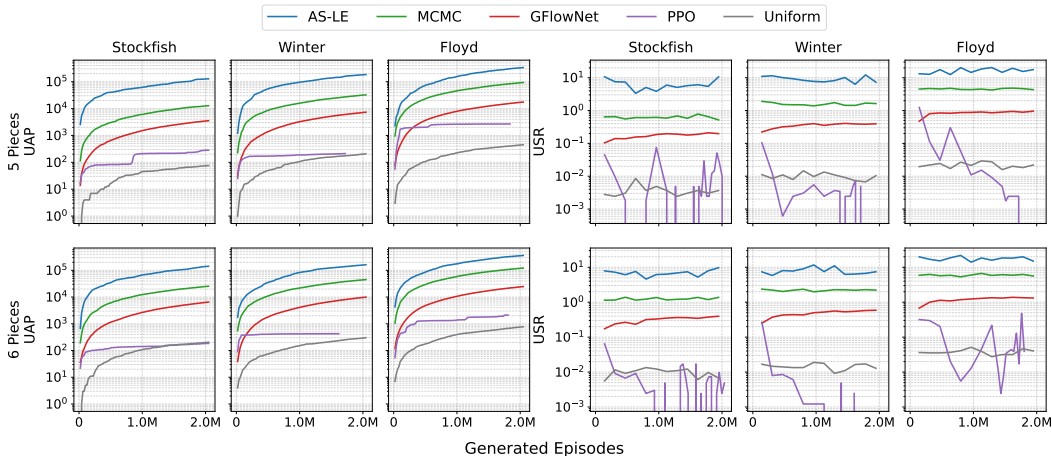

Figure 4: Log-scale sampling method comparison within **ADVCHESS** on three engines with a search-node budget of 6400 (Other search-node budgets are in Appendix C). Unique Adversarial Positions (UAP) and Unique Success Rate (USR) are computed for differing engines and piece counts.

### 6.1 COMPARISON OF SAMPLING APPROACHES

We begin by comparing the effectiveness of the five sampling algorithms integrated into **ADVCHESS**: AS-LE, MCMC, GFlowNet, PPO, and Uniform sampling. Figure 4 shows the number of UAP found and the USR for each method against three chess engines.

The results reveal a stark performance difference between the methods. AS-LE and MCMC dramatically outperform GFlowNet, PPO, and Uniform sampling. As shown in the UAP plots (Figure 4, left), AS-LE consistently discovers orders of magnitude more unique adversarial positions than the model-based and uniform approaches. For example, in the 5-piece setting against Stockfish, AS-LE finds over $10^5$ positions, while GFlowNet and PPO fail to exceed $4 * 10^3$. This underscores the power of AS-LE's design, which leverages the insight that adversarial positions are densely clustered.

The USR plots (Figure 4, right) explain the dynamics behind this performance gap. AS-LE and MCMC maintain a high, stable USR, indicating a consistent and efficient discovery of novel adversarial positions. In contrast, PPO's USR plummets over time, a clear sign of mode collapse where the algorithm repeatedly generates the same few positions. GFlowNet's USR shows a gradual increase, suggesting it is slowly learning relevant features, but its discovery rate remains far too low to be competitive. As expected, Uniform sampling has a flat and extremely low USR, confirming that adversarial positions are too sparse to be found efficiently without a guided search. Given its superior performance, we use AS-LE for all subsequent analyzes.

### 6.2 PERFORMANCE OF **ADVCHESS** WITH AS-LE

Having established AS-LE as the most effective sampling algorithm, we now analyze the performance of the ADVCHESS framework when using AS-LE. Figure 7 illustrates the number of UAP discovered by **ADVCHESS** with AS-LE for each of the three engines across five different search-node budgets, from 400 to 6400 nodes.

**Impact of Engine Strength and Budget.** Figure 7 reveals a clear trend: the stronger the engine and the larger its computational budget, the harder it is to find adversarial positions. Across all search-node budgets, **ADVCHESS** (AS-LE) consistently finds the most positions against Floyd (green line) and

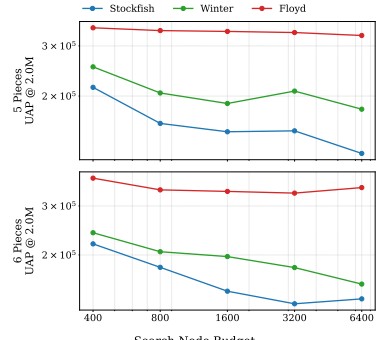

Figure 5: UAP at 2M for **ADVCHESS** with AS-LE for increasing search-nodes.

the fewest against Stockfish (blue line). Furthermore, for the stronger engines (Stockfish and Winter), increasing the search-node budget from 400 to 6400 significantly reduces the number of discovered

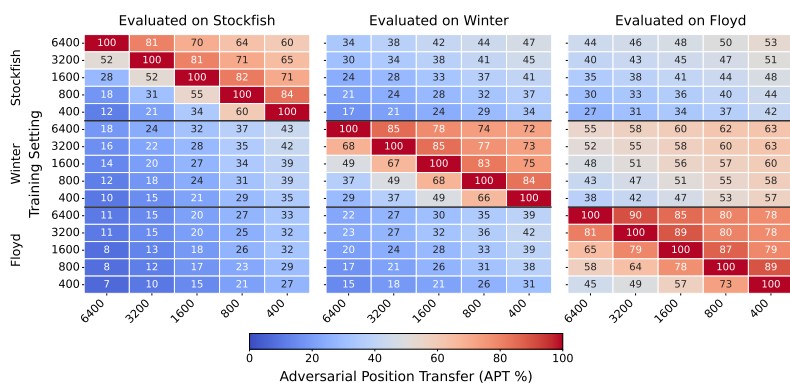

Figure 6: Five piece Inter- and Intra Transferability Matrix

vulnerabilities. This is expected, as a deeper search allows the engine to correct flaws in its initial position evaluation.

**Five vs. Six Pieces.** The trends observed in 5-piece endgames (top row) largely hold for the more complex 6-piece scenarios (bottom row). The relative difficulty of finding adversarial positions across engines and search-node budgets remains consistent, demonstrating the robustness of our approach.

## 6.3 TRANSFERABILITY ANALYSIS

We now examine whether adversarial positions found for one engine setting transfer to others. Figure 6 shows the APT for 5-piece positions, comparing between different engines (inter-engine) and between different search-node budgets for the same engine (intra-engine).

**Inter-Engine Transferability.** We observe a slight trend that adversarial positions tend to transfer from stronger to weaker settings. For instance, positions that fool Stockfish at a high search-node budget are more likely to fool Floyd at a low one than the other way around. This suggests such positions are "universally" difficult. However, transferability is not solely a function of engine strength. Positions generated for Winter transfer more effectively to Floyd than the ones generated for Stockfish. This indicates that our method discovers positions that exploit specific, shared vulnerabilities between Winter's and Floyd's algorithms, which are less prevalent in Stockfish.

**Intra-Engine Transferability.** Within a single engine, transferability decreases as the difference between search-node budgets grows. This holds true whether transferring from a low-to-high or high-to-low budget. When moving to a higher budget, the engine's deeper search can resolve the mistake. Conversely, a lower budget drastically changes the shape of the search tree, which can cause previously adversarial positions to no longer be problematic. Notably, positions generated for Stockfish are the least transferable across its own node settings, suggesting its evaluation is highly sensitive to search depth, making its vulnerabilities more configuration-specific.

## 7 CONCLUSION

We introduced **ADVCHESS**, a framework for the automated discovery of adversarial positions in chess endgames that induce suboptimal play in state-of-the-art engines. Our central contribution stems from the key insight that such positions are not randomly distributed but are, in fact, densely clustered within the vast state space of chess. This understanding enabled the design of a novel and highly effective sampling algorithm, Adversarial Search via Local Exploration (AS-LE), which leverages this clustering to efficiently find vulnerabilities. Through rigorous evaluation against diverse chess engines—Stockfish, Winter, and Floyd—we demonstrated that AS-LE significantly outperforms established methods like MCMC, PPO, and GFlowNets, finding between $2.2\times$ to $10.2\times$ more unique adversarial positions. Our analysis further reveals important characteristics of engine fallibility, including how vulnerabilities transfer across different engines and computational budgets. This work provides developers with a powerful tool to systematically identify and address critical weaknesses, paving the way for more robust and reliable chess engines.

## Ethics Considerations

The primary ethical concern in our work is the potential misuse of adversarial positions to gain unfair advantages in competitive chess environments. While our research aims to improve chess engine robustness, the discovered vulnerabilities could be exploited maliciously. We do not perform studies with live systems, nor do we perform user studies.

**Stakeholders.** We identified several key stakeholders that could be affected by our research - chess engine developers, computer chess tournaments, online chess platforms, and the research community. Chess engine developers risk reputation loss since the reliability of their engines is impacted. The integrity of computer chess competitions could be compromised as adversaries can use our research to gain an unfair advantage. The fair play systems of online chess platforms could be affected. The research community benefits from an improved understanding of chess engine vulnerabilities.

**Mitigating Potential Harm.** Despite potential drawbacks, our analysis of adversarial positions in chess engines is a starting point for developers to improve their chess engines as they see fit. We maintain "Respect for Persons" by engaging transparently with affected stakeholders and considering their interests in our disclosure process. We have followed responsible disclosure practices by promptly informing chess engine developers of our findings, allowing them to address potential weaknesses before public disclosure.

## Reproducibility Statement

To ensure the reproducibility of our findings, we have provided comprehensive details of our methodology and experimental setup. The source code for our **AdvChess** framework, including the from-scratch implementations of our AS-LE algorithm, MCMC, and Uniform sampling, is included in the supplementary materials. The conceptual design of all sampling algorithms and our novel reward function are described in detail in Section 4. Our full experimental setup, including the specific versions of the three chess engines used (Stockfish, Winter, and Floyd), their query configurations, and the evaluation metrics, is specified in Section 5 . Furthermore, all hyperparameters required to reproduce our results with the learning-based samplers (GFlowNet and PPO), such as learning rates, batch sizes, and reward component weights, are provided in Appendix B, with a comprehensive summary in Table 2. The implementation relies on PyTorch, TorchGFN, and stable-baselines3, as listed in Appendix B. An environment.yaml file is included with the code, as well as a readme file to help researchers install the necessary engines (including versions) and tablebase files.

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
