# OpenReview forum: "Discovery of Adversarial Endgame Chess Positions"
_ICLR.cc/2026/Conference — Submitted to ICLR 2026_

### Official Review · Reviewer_beaU · 2025-10-25

**Soundness:** 3
**Presentation:** 3
**Contribution:** 3
**Rating:** 6
**Confidence:** 4

**Summary:**

This paper introduces ADVCHESS, a novel framework to automatically generate adversarial chess positions. The authors focus on identifying fair and legal positions where engine failures result in significant outcome changes. In addition, the authors design a reward function AS-LE to simultaneously encode three desired aspects: (1) adversarial score, (2) position legality, and (3) material balance of the position, for guiding the generative process toward the final objective.

Main contributions of the authors include:
1. Benchmarking several advanced sampling methods (MCMC, PPO, and GFlowNets) against
the uniform baseline, and analyze their performance and limitations in adversarial position discovery.
2. Designing and implementing a novel search algorithm, AS-LE, as the core sampling component within the ADVCHESS framework, which combines local and uniform search and significantly outperforms all tested baselines.

**Strengths:**

In terms of significance, the paper addresses the systematic generation and analysis of adversarial positions: specialized board configurations that reveal weaknesses in engine decision-making, leading to suboptimal play and altering theoretical game outcomes from wins to draws or losses. The discussion on sampling methods is mostly well done.
In terms of originality, their main contribution is designing a novel search algorithm, Adversarial Search via Local Exploration (AS-LE), which leverages a combination of local and uniform search to effectively explore the vast space of chess positions and integrates it with the ADVCHESS framework to find rare, failure-inducing endgame positions by navigating the vast and sparse search space. The authors present and perform a clear empirical study to evaluate their framework by benchmarking it against three diverse chess engines (Stockfish, Winter, and Floyd) with three different metrics. They also compare their proposed sampling algorithm, AS-LE, against four other methods to establish its superior efficiency.
Lastly, the quality of this work is analyzing the transferability of the discovered positions across various engine configurations, as shown in Figure 6.

**Weaknesses:**

Overall, this paper could be a significant algorithmic contribution, with the caveat for some clarifications on the theory and experiments. Given these clarifications in an author response, I would be willing to increase the score.

For the theory–sampling algorithm adaptation section 4.1, there are a few steps that need clarification and further clarification on novelty. Did you use MCMC, PPO, and GFlownet as base algorithms and integrate those with the new reward function? So your reward seems just a component and an integration to these algorithms, rather than designing a new sampling algorithm? In other words, it is not clear if AS-LE is a new sampling algorithm, or it is just an integration of already existing sampling algorithms? How did you manage to overcome challenge 2 (Complexity of Strategies (C2))? Can your reward function alone circumvent all 4 challenges?

For the experiments, on page 7, you mentioned that “Further implementation details and algorithm
hyperparameters are given in the Appendix B.”, but no appendix is provided. More about implementation details would be illuminating. There is also another missing appendix. In Figure 4, “(Other search-node budgets are in Appendix C)”. It would be absolutely good to see more details regarding these experiments.

**Questions:**

For the experiments, the following should be addressed:
1. It would have been better to also show the performance graphs with and without the improvements to the reward function.
2. The central contribution is designing a reward function. It would be beneficial to see, empirically, how the performance differs with and without the reward.
3. Implementation details, hyperparameters, and parameters are missing. I would like to know how changing the parameters affects the performance?

---

> ### Author Response · Authors · 2025-11-22
> **Response for Reviewer beaU**
>
> We thank the reviewer for the positive assessment and for identifying the AS-LE algorithm as a key contribution. We apologize for the confusion regarding the Appendix. The full paper version with the Appendix can be found in the supplementary materials.
>
> ## Q1. Missing Appendix and Implementation Details
> Hyperparameters: Table 2 (Page 13 of supplementary) details learning rates (3x10^-4), batch sizes (256), and reward weights. Appendix C contains the additional experiments (Pages 13-15).
>
> ## Q2. Regarding AS-LE
> AS-LE is a distinct search algorithm, not just a reward wrapper. MCMC/PPO/GFlowNet uses the reward function to update a probability distribution or policy. AS-LE explicitly switches between Global Random Search (to find a seed) and Local Depth-First Search (to mine the cluster). This algorithmic structure (switching modes based on hit rate) is the novelty that allows it to outperform the generative models by orders of magnitude (as shown in Figure 4).
>
> ## Q3. Reward Ablation
> Please see Table R1 in the General Response. The data confirms that the reward components (specifically Material Balance) are necessary for the generative model to locate valid adversarial positions.

---

### Official Review · Reviewer_N7Jf · 2025-10-28

**Soundness:** 2
**Presentation:** 4
**Contribution:** 2
**Rating:** 4
**Confidence:** 4

**Summary:**

The paper aims to discover the adversarial chess positions that will let chess engines make mistakes, specifically in the endgame. To discover these positions, the paper developed the ADVCHESS framework, which uses the sample methods to generate adversarial chess positions. The authors first compare several sample methods, including uniform sampling, MCMC, GFlowNets, and PPO, against a custom reward function. Then, they observe that adversarial chess positions tend to cluster densely in the state space, leading to the main method, Adversarial Search via Local Exploration (AS-LE). AS-LE operates in two phases: a Discovery phase that performs via uniform random search until an adversarial position is found, and an Exploration phase that performs via DFS. The results show that AS-LE outperforms other sample methods, collecting the maximum number of unique adversarial chess positions.

**Strengths:**

1. The paper is well-structured and well-written. The definition of adversarial positions and the reward function are clearly addressed. Figures (e.g., Figures 2-3) and their explanations improve readability, allowing the readers to understand easily.

2. The research topic of finding adversarial chess positions is underexplored. The paper proposes a novel framework, ADVCHESS, which includes all the sample methods mentioned above, to generate adversarial chess positions. Especially the outperform sample method AS-LE, which is easy and powerful.

3. The success of the ADVCHESS framework pioneers the field of systematically generating adversarial chess positions, enabling developers to identify and mitigate engine weaknesses more effectively.

**Weaknesses:**

Overall, the results are interesting, but the motivation of this paper is not clear. I think the authors should carefully address each of the following weaknesses.

1. For example, this paper focuses on using endgame positions to identify adversarial positions. However, it is not clear why it only investigates endgame positions, which are relatively easy positions. Furthermore, it is also not clear how these discovered endgame positions help.

2. Although chess has an extensive endgame database, the proposed method should not be limited to endgame scenarios and the chess environment. It would be convincing to incorporate other games (e.g., Go, Hex, Othello) to demonstrate the generalizability.

3. Studies on adversarial in board games have been widely investigated, and some of them can also generate the adversarial position or policy [1-2]. Their approach can also automatically discover adversarial positions without the limitation to endgame positions. The paper should carefully review these backgrounds, explicitly describe the differences between the proposed methods and previous methods, and even compare the approaches in the experiments.

[1] Lan, Li-Cheng, et al. "Are alphazero-like agents robust to adversarial perturbations?." Advances in Neural Information Processing Systems 35 (2022): 11229-11240.

[2] Wang, Tony Tong, et al. "Adversarial policies beat superhuman Go AIs." International Conference on Machine Learning. PMLR, 2023.

4. The paper lacks an analysis of the number of different clusters identified by each sampling method. The experiment results indicate that the Uniform Sampling method is underperforming, while AS-LE is outperforming. This suggests that the improvement of AS-LE is limited to similar positions in the same cluster, rather than increasing the number of clusters. A comparison of cluster diversity across sampling methods would be more convincing.

**Questions:**

1. Please address each concern raised in the weaknesses.

2. What's the motivation of this work? Why is this work mainly focusing on endgames? Can it be applied to other games without the endgame database? How can these discovered endgame positions benefit future research?

3. The part of "Intra-Engine Transferability" and results show that transferability decreases as the difference between search-node budgets grows. Intuitively, the adversarial chess positions that fool a stronger engine (high simulation) should also mislead a weaker one (low simulation), implying high transfer from weak to strong. However, the results show the opposite trend. Providing a deeper analysis would be helpful.

4.  Section 6.2 refers to Figure 7, but no Figure 7 appears in the paper. Does it mean Figure 5, or is a figure missing?

5. Figure 6 lacks of caption to explain how to read the experiment results, which makes readers hard to understand.

---

> ### Author Response · Authors · 2025-11-22
> **Response for Reviewer N7Jf**
>
> We thank the reviewer for the excellent score on presentation and for highlighting the importance of this underexplored topic.
>
> ## Q1. Missing Figure 7 and Figure 6 Caption
> It is indeed a typo and should read Figure 5; however, Figure 7 does exist in the Appendix on page 13. The full paper with appendix can be found in the supplementary materials. We apologize for the confusion!
>
> Thank you for the constructive criticism. We will improve the Figure 6 caption to state as follows: “Five-piece Adversarial Position Transferability (APT) matrix. The results show that intra-engine transferability decreases as the gap in search-node budgets widens. Inter-engine results reveal that transferability is not solely a function of strength; notably, positions generated for Winter transfer more effectively to Floyd than those from the stronger Stockfish, suggesting shared algorithmic vulnerabilities.”
>
>
> ## Q2. Motivation: Why Endgames?
> We chose endgames specifically to ensure scientific rigor.
>
> In endgames, we have Tablebases (perfect play). This means an adversarial position is a proven error. In Go/Hex or Chess middlegames, we only have "superhuman estimates." If AlphaZero suggests a move, it is not a proven fact. At best, we can approach the ground truth by consulting multiple superhuman AIs or by increasing the target engine’s compute budget when acting as a judge.
>
> Comparison to [1, 2]: Previous works like Wang et al. (2023) generate adversarial policies to beat a specific victim. Our work generates positions where the victim fails against optimal play. This is a fundamental difference: we are debugging the engine's understanding of truth, not just beating it. Furthermore, while these methods are successful in Go, we are unaware of whether these methods work in Chess, which has a more constrained search space.
>
> ## Q3. Intra-Engine Transferability
> Transfer from High-to-Low is still much higher than low to high. For example, transferability from 6400 nodes to 400 nodes is still 60%, compared to only 12% in the opposite direction.
>
> **Low-to-High Transfer**: A position that fools Stockfish at 400 nodes (shallow search) is often solved correctly at 6400 nodes (deep search). The deeper search resolves the horizon effect or tactical oversight. Thus, transfer is low.
>
> **High-to-Low Transfer**: A position that fools Stockfish at 6400 nodes is likely more complex (a "deep" trap). A version running at 400 nodes might not even enter the branch of the search tree that causes the failure. Thus, the specific adversarial nature of the position does not transfer.
>
> ## Q4. Cluster Diversity
> This is an excellent point. While we measure "Unique Adversarial Positions" (UAP), AS-LE specifically targets local clusters. However, the Discovery Phase of AS-LE (uniform random jumps) ensures we find different clusters. Once a new cluster is found, the Exploration Phase mines it. This balance ensures both diversity (via Discovery) and quantity (via Exploration).

---

> > ### Comment · Reviewer_N7Jf · 2025-11-26
> >
> > Thank you for your response. However, the clarifications are still not satisfactory.
> >
> > 1. I remain unconvinced why using endgame is important. The goal of this paper is to identify positions where engine failures result in significant outcome changes. Under this objective, it is unnecessary to use only the endgame position with ground truth. Many previous approaches can identify failure position without ground truth. Since ground truth is unavailable outside endgames, this limitation also limits the proposed approach to endgame positions only.
> >
> > 2. As the current approach depends on endgame, it is questionable to generalize the proposed method to other games without endgame database, such as Go, Hex, and Othello. Including an additional experiment on another game would help demonstrate that the approach can generalize.
> >
> > 3. Regarding cluster diversity, do you have any statistical results comparing the diversity achieved by each approach?

---

> > > ### Author Response · Authors · 2025-12-03
> > > **New Clustering Data & Scope Clarification**
> > >
> > > We want to share one final set of experimental results to address your questions regarding clustering statistics and the diversity of our sampling.
> > >
> > > ### 1. Quantitative Evidence of Clustering
> > > To validate our claim that adversarial positions exist in dense clusters, we measured the "adversarial density" around 800 random regular (non-adversarial) positions versus 800 adversarial positions for Stockfish. We computed the probability that a random legal neighbor (perturbation moving an arbitrary piece to a new square) is adversarial.
> > >
> > > | Piece Count | Node Budget | Density around Regular Pos. | Density around Adv. Pos. | Ratio (Clustering Factor) |
> > > | :--- | :--- | :--- | :--- | :--- |
> > > | **5** | **6400** | 0.11% | 5.39% | **49x** |
> > > | **5** | **1600** | 0.24% | 6.86% | **29x** |
> > > | **5** | **400** | 0.53% | 10.59% | **20x** |
> > > | **6** | **6400** | 0.20% | 6.04% | **30x** |
> > > | **6** | **1600** | 0.39% | 7.73% | **20x** |
> > > | **6** | **400** | 0.77% | 11.56% | **15x** |
> > >
> > > The data confirms the topology of the search space: finding an adversarial position near another one is 15x to 49x more likely than finding one by chance.
> > >
> > > This dictates why AS-LE works and ensures diversity. The Discovery Phase (uniform sampling) is necessary to find the distinct, hard-to-hit "seeds" (the 0.11% case) scattered across the manifold. The Exploration Phase is then necessary to mine the pocket (the 5.39% case). If we only used uniform sampling, we would miss the clusters; if we only used local search, we would lack diversity. AS-LE maximizes both.
> > >
> > > ### 2. The Necessity of Endgames (Ground Truth)
> > >
> > > We insist on focusing on endgames because they are the only chess domain where Ground Truth exists. In a middlegame, an "adversarial" label is merely a disagreement between two engines (e.g., Stockfish vs. AlphaZero). We cannot prove which one is right. In endgames, via Tablebases, we have a mathematical oracle. If Stockfish evaluates a provable "Loss" as a "Draw," it is objectively broken. This rigor is critical for debugging engine understanding, which distinguishes our work from previous studies like Wang et al. that aim to beat an agent.
> > >
> > > ### 3. Generalization
> > >
> > > Implementing valid oracles for Go, Hex, or Othello is a significant engineering undertaking outside the scope of this work. We selected Chess to rigorously benchmark the AdvChess framework. The clustering data above relies on the properties of discrete minimax search spaces, suggesting the methodology is transferable, but we maintain Chess as the primary testbed for this study.

---

### Official Review · Reviewer_aYhL · 2025-10-30

**Soundness:** 2
**Presentation:** 3
**Contribution:** 1
**Rating:** 2
**Confidence:** 3

**Summary:**

This paper introduces an algorithm for constructing endgame Chess positions for which engines deviate from optimal play. While previous studies rely on uniform sampling methods for position generation, AdvChess generates positions systematically through heuristics, endgame tablebases, and engine outputs. The authors compare against several baselines and determine that their method discovers many more adversarial samples.

**Strengths:**

Systematic methods for finding failure cases could be important for driving further engine improvements. Given the authors' claim that prior work generates these positions uniformly, it seems important to consider new methods.

The presentation of the paper is strong, and it is generally well-written. Empirical and algorithmic details are explained precisely.  The provided code and hyperparameter settings are much appreciated and contribute positively to the quality of the work.

**Weaknesses:**

**Baselines.** The baselines provided in this paper seem somewhat arbitrary rather than from prior work, and their choices are under-justified. Why should I expect the particular instance of MCMC to be effective at generating such positions? Was PPO tuned effectively for this particular task? Choices for those two algorithms could make a significant difference in terms of the number of adversarial positions generated.

**Significance of the contribution** The paper's main contribution is AdvChess, which performs well compared to the tested baselines, but I'm unconvinced about AdvChess' relevance outside of that and the transferability hypotheses. Could AS-LE be extended to do the same thing in other perfect information games? In general, more discussion about the limitations of the work would benefit the paper significantly.

**Relevance to the ICLR community** It is not clear to me that this is an appropriate venue for this kind of work. The choice of "primary area" for the submission is perhaps evidence of this, as it seems like a stretch to classify this as a contribution to *probabilistic methods*.

**Questions:**

See weaknesses.

---

> ### Author Response · Authors · 2025-11-22
> **Response for Reviewer aYhL**
>
> We appreciate the reviewer’s feedback. We believe in the relevance of our work to ICLR and are happy to provide justification of baselines below.
>
> ## Q1. Relevance to ICLR
> ICLR has a strong tradition of publishing work at the intersection of AI for Games (from this year, I particularly enjoyed Human-Aligned Chess With a Bit of Search by Yiming Zhang et. al), Robustness, and Generative Modeling. Our work contributes to:
>
>
> 1. AI Robustness: Systematically finding failure modes in superhuman systems (Stockfish).
> 2. Generative Sampling: Demonstrating that SOTA methods (GFlowNet, PPO) fail in domains with sparse rewards and dense solution clusters, and proposing a search-based alternative (AS-LE) that solves this. This is a significant insight for the generative modeling community.
>
> ## Q2. Relevance of Baselines
> Regarding MCMC, we tried several versions and kept the best based on preliminary results. The one in the paper is the only one that outperformed the GFlowNet baseline.
>
> We selected these baselines because they represent the standard approaches for black-box optimization and generative sampling:
> * **Uniform Sampling**: The standard used in prior chess literature (Sadmine et al., 2023).
> * **PPO (RL)**: The standard for optimizing non-differentiable objectives.
> * **GFlowNets**: The current SOTA for generating diverse samples from a distribution (Bengio et al., 2021).
> * **MCMC**:  A classical baseline for sampling from energy-based models.
>
> The failure of these "standard" methods is exactly what justifies the design of AS-LE.
>
> ## Q3. Significance Outside of Chess
> While demonstrated in Chess, the core insight, that adversarial examples often exist in dense local clusters ("pockets"), is relevant to other discrete domains, such as finding jailbreak prompts for LLMs or adversarial patches in images. AS-LE provides a template for exploiting this clustering in other perfect information games.

---

### Official Review · Reviewer_a7oD · 2025-11-01

**Soundness:** 2
**Presentation:** 3
**Contribution:** 1
**Rating:** 4
**Confidence:** 2

**Summary:**

This paper introduces ADVCHESS, a novel framework for the systematic discovery of adversarial endgame positions in chess—board configurations where state-of-the-art engines (e.g., Stockfish) deviate from theoretically optimal play as verified by endgame tablebases (EGTBs).

**Strengths:**

**Novel and Impactful Problem Formulation**:
The paper addresses a timely and underexplored problem: the systematic discovery of adversarial endgame positions that expose failures in state-of-the-art chess engines. Despite the widespread trust in engines like Stockfish, their vulnerabilities in theoretically solvable endgames have not been rigorously studied through automated generation—this work fills that gap.

**Weaknesses:**

**Lack of Theoretical Analysis**: The paper observes that adversarial positions are “densely clustered” and leverages this empirically to design AS-LE, but it does not provide a theoretical explanation for why this clustering occurs or how it relates to engine internals (e.g., evaluation heuristics or search pruning).

**Limited Generalizability of AS-LE**: The proposed AS-LE algorithm relies on local perturbations (e.g., moving or transforming a single piece), which is effective in sparse endgame settings but may not scale to midgame or more complex positions with higher-dimensional state spaces.

**Insufficient Connection to Prior Work**: While the paper cites related studies, it does not deeply analyze whether the discovered adversarial positions represent new types of engine failures or merely reproduce known weaknesses.

**No Ablation on Reward Components**: The reward function combines adversarial score, legality, and material balance, but the paper does not include ablation studies to assess the contribution of each component to sampling efficiency or diversity.

**Questions:**

**On the underlying mechanism of clustering**:
   The paper notes that adversarial endgame positions are “densely clustered” in the state space and leverages this observation to design the AS-LE algorithm. However, the **root cause of this clustering remains unclear**. Have the authors investigated whether this phenomenon is linked to specific weaknesses in engine evaluation functions (e.g., misjudgment of certain tactical motifs) or search-pruning strategies (e.g., premature pruning of critical branches in alpha-beta search)? Could the authors provide deeper attribution analysis or visual evidence to support this insight?

**On scalability and generalizability of the method**:
   AS-LE relies on local perturbations—such as moving or transforming a single piece—which works well in 5–6 piece endgames where ground-truth labels are available via endgame tablebases. Have the authors considered—or do they plan to extend—the ADVCHESS framework to **middlegame scenarios** (e.g., positions with 8–10 pieces)? In settings without perfect ground truth, how would “adversariality” be defined or approximated? Would the local exploration strategy of AS-LE still be effective in such higher-dimensional and less-structured state spaces?

---

> ### Author Response · Authors · 2025-11-22
> **Response for Reviewer a7oD**
>
> We thank the reviewer for finding our problem formulation "novel and impactful" and our presentation "good." We address the specific concerns below.
>
> ## Q1. Why are adversarial positions densely clustered? Is it linked to evaluation heuristics?
> In the general case, it is difficult to prove exactly what is going on due to the complex interactions between engine search and evaluation. However, we believe the clustering arises from the "slack" inherent in winning chess positions.
>
> That is, if a position P is a "Win" (value +1) and the engine evaluates it as a "Draw" (value 0), moving a non-critical piece (e.g., a pawn far from the action) to an adjacent square usually results in a position P’ that preserves the theoretical "Win" status. This is likely due to the feature maps in the evaluation functions being almost identical if there is only a slight change in the position.
>
> We believe both examples in Figure 1 occurred due to issues in the evaluation function. In practice, we identified other examples where errors occurred due to the horizon effect or overly aggressive pruning. We will add a case study to the Appendix to further the discourse on this complex issue.
>
> ## Q2. Can this extend to middlegames or 8-10 pieces?
> While AS-LE's local perturbation strategy is transferable to middlegames, the definition of "adversarial" becomes the bottleneck (as noted in the General Response). In a middlegame, we would need to substitute the Tablebase (Ground Truth) with a "Stronger Oracle" (e.g., an engine given 100x more compute time). We believe applying AdvChess to middlegames using "Compute-As-Truth" is a promising future direction, but we restricted this work to endgames to maintain mathematical rigor in our error definitions.
>
> ## Q3. Reward Ablations
> Please see Table R1 in the General Response. The data confirms that the Material Balance reward is particularly essential for guiding the search toward valid, high-quality adversarial pockets.

---

### Author Response · Authors · 2025-11-22
**General Response: Location of Appendix, New Ablation Results, and Scope**

We thank the reviewers for their constructive feedback and are encouraged that they found the problem "novel and impactful" (a7oD) and the framework "well-structured" (beaU).

## Location of Appendices and Missing Figures
We apologize for the confusion regarding the location of the implementation details. The Appendix was submitted as a file in the Supplementary Material (*ICLR_Adversarial_Chess_Endgame_Positions_With_Supplementary.pdf*).

* **Hyperparameters:** Full details (learning rates, batch sizes, reward weights) are in Table 2 (Page 13 of Supplementary).
* **Additional Experiments:** The additional sampling experiments are in Appendix C, pages 13-15 of the Supplementary version.
* **Code:** Source code for AdvChess, including sampling baselines, is also included in the supplementary material.

**Action:** We will add portions of the Appendix that were asked about by reviewers to the main paper for the camera-ready version (which has a higher page limit). We will move critical figures into the main text in the camera-ready version to ensure better flow.

## New Results: Reward Function Ablation (Response to a7oD, beaU)
Reviewers a7oD and beaU asked for an ablation study to justify the reward components. We present **Table R1** below, which ablates the reward components for 5-piece positions targeting Stockfish using GFlowNet sampling after 5 million generated samples.

| Nodes | Full Reward | No Valid Pos. | No Mat. Bal. | No Adv. Outcome |
| :--- | :--- | :--- | :--- | :--- |
| **400** | **52,743** | 47,649 | 30,919 | 50,379 |
| **800** | **36,695** | 33,049 | 21,883 | 35,877 |
| **1600** | **24,549** | 21,600 | 14,509 | 24,037 |
| **3200** | **15,726** | 13,575 | 9,434 | 15,463 |
| **6400** | **9,820** | 8,308 | 6,060 | 9,538 |

* **Material Balance ($\beta_m$):** Removing this reward causes a massive performance drop (e.g., -41% at 400 nodes). This confirms that enforcing realistic material configurations is the most critical constraint for discovering valid adversarial candidates.
* **Valid Position ($\beta_l$):** Removing this leads to a moderate drop. While the model learns valid moves relatively quickly, the explicit reward stabilizes training.
* **Adversarial Outcome ($\beta_o$):** The impact is present but smaller. This is expected because learning the exact adversarial distribution of a superhuman engine is an extremely sparse signal for a global sampler; this difficulty is precisely why our proposed AS-LE (which adds local search) outperforms these global baselines.

## Why Endgames? (Response to a7oD, N7Jf)
Reviewers a7oD and N7Jf asked about the focus on endgames. We focus on endgames because they are the only chess domain where perfect, theoretical Ground Truth exists (via Endgame Tablebases).

In middlegames, no oracle exists. If our system claims a position is adversarial because Stockfish played Move A but AlphaZero prefers Move B, we cannot prove Stockfish is “wrong”; only that it disagrees with another engine.

In endgames, we have mathematical proof of the game’s outcome. So, even if Stockfish claims a position is still a draw after a losing move, we know for sure what the truth is.

---

### Meta-Review · Area_Chair_iaBF · 2026-01-14

**Summary:**

This paper proposes ADVCHESS, a framework for systematically generating (sampling) chess endgame positions where a target engine’s chosen move changes the outcome relative to endgame tablebases.

While the paper is clearly written and the experimental results support the claim that the proposed method (AS-LE) is effective within the ADVCHESS pipeline for endgame-tablebase-defined adversarial positions, the core weaknesses raised by the reviewer is that the contribution is perceived as incremental by multiple reviewers (especially relative to “search + local exploration” as a general idea), with limited theoretical or mechanistic analysis of the reason why clustering occurs, and limited evidence that the approach (or key insights) generalize beyond tablebase solvable endgames.​

**Reviewer Concerns:**

In the rebuttal, the authors clarified that implementation details and appendices were in the supplementary materials, addressed some writing confusion, and added a reward-component ablation (Table R1). The response also offered a plausible explanation for clustering (“slack” in winning positions and evaluation similarity under small perturbations), plus additional quantitative clustering statistics for justification of the proposed method.

However, several high-level concerns remain outstanding and continue to weigh against acceptance. The rebuttal does not fully resolve doubts about broader significance and generality, especially whether the main insight extends beyond the special setting where an oracle (tablebase) provides ground truth labels, and whether the contribution is primarily a domain-specific engineering study rather than an top-tier ML conference level methodological advance.

**Reviewer Scores:**

Reviewer beaU (score 6) explicitly stated willingness to increase their score given adequate clarifications. Reviewer N7Jf (rating 4) participated in follow-up discussion expressing continued dissatisfaction, stating "the clarifications are still not satisfactory" and reiterating concerns about endgame dependence and lack of generalization experiments, suggesting they would likely maintain their rating. Reviewer a7oD (rating 4) requested specific analyses that were partially addressed through the ablation study, suggesting they might slightly increase their borderline score. Reviewer aYhL gave the lowest score of 2 (reject) based on fundamental concerns about venue appropriateness and contribution significance that were not fully resolved by the author response, so aYhL might not change their score to support this paper.

---

### Decision · Program_Chairs · 2026-01-26

Reject